# The impact of Arctic sea ice loss on mid-Holocene climate

Hyo-Seok Park[1], Seong-Joong Kim[2], Kyong-Hwan Seo[3], Andrew L. Stewart ⓘ [4], Seo-Yeon Kim[5] & Seok-Woo Son[5]

Mid-Holocene climate was characterized by strong summer solar heating that decreased Arctic sea ice cover. Motivated by recent studies identifying Arctic sea ice loss as a key driver of future climate change, we separate the influences of Arctic sea ice loss on mid-Holocene climate. By performing idealized climate model perturbation experiments, we show that Arctic sea ice loss causes zonally asymmetric surface temperature responses especially in winter: sea ice loss warms North America and the North Pacific, which would otherwise be much colder due to weaker winter insolation. In contrast, over East Asia, sea ice loss slightly decreases the temperature in early winter. These temperature responses are associated with the weakening of mid-high latitude westerlies and polar stratospheric warming. Sea ice loss also weakens the Atlantic meridional overturning circulation, although this weakening signal diminishes after 150–200 years of model integration. These results suggest that mid-Holocene climate changes should be interpreted in terms of both Arctic sea ice cover and insolation forcing.

[1] Korea Institute of Geoscience and Mineral Resources, Daejeon 34132, South Korea. [2] Korea Polar Research Institute, Incheon 21990, South Korea. [3] Department of Atmospheric Sciences, Pusan National University, Busan 46241, South Korea. [4] Department of Atmospheric and Oceanic Sciences, University of California, Los Angeles, CA 90095-1565, USA. [5] School of Earth and Environmental Sciences, Seoul National University, Seoul 08826, South Korea. Correspondence and requests for materials should be addressed to H.-S.P. (email: hspark1@gmail.com)

Northern Hemisphere summer solar insolation was anomalously strong during the early-mid Holocene, causing the Holocene thermal maximum from around 9000 years to 5000 years before present (BP)[1,2]. The thermal maximum was particularly pronounced at high latitudes including Greenland, the Canadian Arctic, and Alaska[3–6]. Proxy data also indicate that Arctic sea ice cover was smaller than the present during the mid-Holocene (around 6 ka BP) over Fram Strait, Baffin Bay, and the Labrador Sea[7]. This is because summer insolation can warm the Arctic rapidly throughout the season, as sea ice decline can amplify the warming via various feedbacks, referred to as polar amplification[8–10]. These paleo-proxy records are supported by climate models that simulate substantial reduction of the summer and autumn sea ice cover during the mid-Holocene relative to the pre-industrial climate[11,12].

Because Arctic sea ice influences surface albedo and ocean–atmosphere heat exchange, as well as ocean's freshwater content[13], we anticipate that Arctic sea ice decline may have contributed substantially to mid-Holocene climate change. Indeed, modeling studies associate the recent and future declines of Arctic sea ice cover with significant impacts on Northern Hemisphere weather patterns[14–21]. In particular, polar stratospheric warming, a negative Arctic Oscillation (AO)-like weakening of mid-high latitude westerlies, and a slight cooling over Central–East Asia appear robustly in the climate model simulations. In contrast, there has been little research on how Arctic sea ice cover contributed to mid-Holocene climate change.

In this study, we show that the Arctic sea ice decline had profound effects on mid-Holocene climate throughout the year. Specifically, we present modeling evidence that the impact of sea ice decline on mid-latitude temperature might be as large as that of the direct effect of insolation. We perform three simulations to separate the effect of sea ice loss from that of direct insolation: one with pre-Industrial insolation and sea ice, one with mid-Holocene insolation and sea ice, and one with mid-Holocene insolation and pre-Industrial sea ice (see Methods). We use a state-of-the-art climate model, the Community Earth System Model, version 1 (CESM1) with the Community Atmospheric Model, version 5 (CESM1–CAM5)[22]. The root-mean-square errors of sea ice extent and volume between CESM1–CAM5 and observations are one of the lowest[23] among 49 climate models that have participated in phase 5 of the Coupled Model Intercomparison Project (CMIP5).

## Results

**Seasonal changes in surface air temperature.** In our mid-Holocene simulation, Arctic sea ice concentration (SIC) in summer-autumn decreases by 30–35% over wide areas of the Arctic relative to the pre-industrial climate (Fig. 1), which is consistent with the multi-model averaged SIC anomalies in the mid-Holocene[11,12]. Figure 2a shows the annual-mean response of mid-Holocene surface air temperature (SAT) to Arctic sea ice loss. The substantial warming over and around Greenland, which also appears in proxy data[4], has previously been reported as a response to Arctic sea ice loss[24,25]. The impact of Arctic sea ice loss is not limited to high latitudes: sea ice loss warms the North Pacific Ocean and North America, but slightly cools the North Atlantic Ocean. The longitude–time Hovmöller plot of SAT averaged between 45°N and 62°N (Fig. 2c) shows that the warming over the Pacific and North America persists throughout the season, whereas cooling over Central and East Asia appears in winter, especially in December and January. The amplitudes of these ice melting-induced annual-mean SAT anomalies (Fig. 2a) are comparable to those of the direct insolation forcing (Fig. 2b). The insolation forcing warms the North Atlantic and Europe, and

cools North America. The warming over Europe is usually driven by anomalously strong summer insolation forcing[26] and this is supported by multiple pollen records[27]. Indeed, the Hovmöller plot (Fig. 2d) verifies that the mid-Holocene insolation forcing causes strong summer warming and winter–spring cooling over the entire longitude circle. While the seasonal impact of insolation forcing on SAT in the mid-latitudes is about twice as large as that of sea ice loss (compare Fig. 2c, d), the annual-mean SAT anomalies driven by insolation forcing and sea ice loss are comparable with each other (compare Fig. 2a, b).

**The wintertime SAT.** The influence of Arctic sea ice loss on Northern Hemisphere climate is particularly pronounced during winter, from December to March (Fig. 3a): SAT over the Eurasian continent, specifically Central and East Asia, decreases by ~0.6 K. This result is comparable to model-simulated eastern Eurasian continent cooling due to the recent sea ice decline[19,20]. This Eurasian continent cooling occurs mostly in early winter (December–January), during which SAT decreases by 0.8 K over wide areas of Central and East Asia (Supplementary Fig. 2). The decrease in SAT over the Eurasian continent is associated with the development of anomalously high sea level pressure (SLP) in Arctic and sub-Arctic Russia and the deepening of the Aleutian low (Supplementary Fig. 1), consistent with a recent projection of Arctic sea ice loss[21] forced by Representative Concentration Pathway (RCP) 8.5. The development of high pressure in sub-Arctic Russia often causes cold eastern Eurasian winters[15,28,29]. Over Central and East Asia, both sea ice loss and insolation forcing act to decrease temperature (Fig. 3a, b), exacerbating the cold continental winter. These model results are supported by proxy records that the East Asian winter monsoon was probably stronger in early- and mid-Holocene than the present[30].

Over North America, the anomalous winter insolation decreases SAT by more than 2 K (Fig. 3b), which is not surprising given that the winter insolation weakens by 5–10 W m$^{-2}$ and land has a relatively low heat capacity. On the contrary, the sea ice loss increases SAT by 2 K, especially over Alaska and Canada (Fig. 3a). This is partly because of the Aleutian low strengthening (Supplementary Fig. 1), which is followed by the northward transport of warm southern air to Alaska and western Canada[31], increasing SAT over these regions. This wintertime surface warming over North America also appears in a climate model simulation of a projected Arctic sea ice loss, forced by RCP 8.5 with a well-resolved stratosphere[19]. This suggests that without the effect of sea ice loss, the weakening of winter insolation would have led to harsher winters over North America during the mid-Holocene. The summed effects of insolation forcing and sea ice loss on SAT manifest as a warm Arctic and cold continent pattern in winter, consistent with the Paleoclimate Modelling Intercomparison Project phase 3 (PMIP3) multi-model average (Supplementary Fig. 3). The wintertime Arctic warming in the PMIP3 multi-model average is generally weaker than that in CESM1, though the 4 warmest of the PMIP3 models simulate Arctic warming that is comparable to CESM1 (Supplementary Fig. 3).

**The wintertime atmospheric circulations.** How does Arctic sea ice loss affect the mid-latitude atmospheric circulation pattern during winter? While variability of the atmospheric circulation obscures the mid-latitude climate response to the recent Arctic sea ice loss[32], the zonal-mean circulation response to the recent and the projected Arctic sea ice loss is more robust[33]. In particular, Arctic sea ice loss causes a negative AO-like circulation pattern[16–21]. Consistent with these studies, Arctic sea ice loss in our mid-Holocene simulation substantially weakens the mid-high

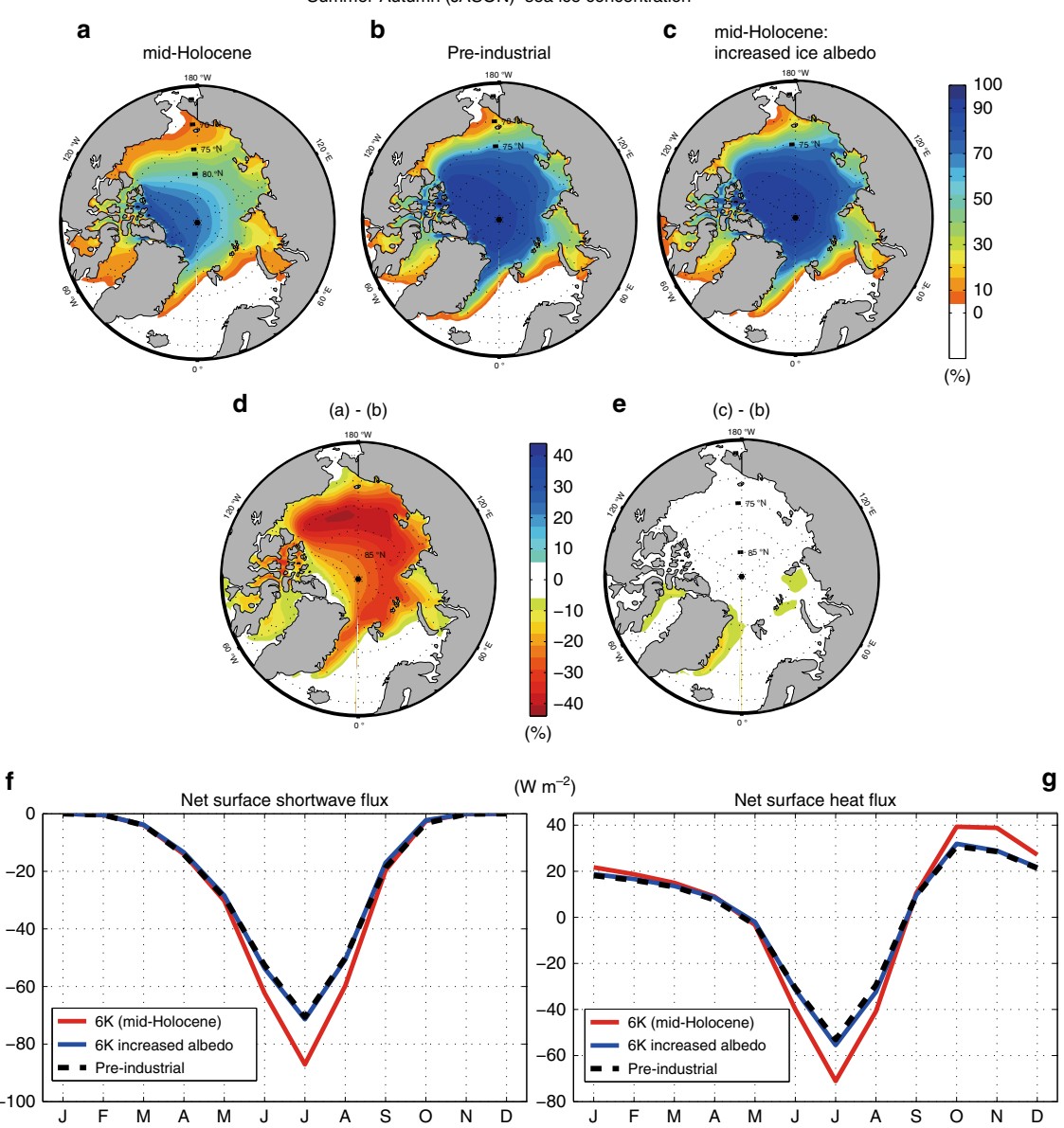

**Fig. 1** Simulated Arctic sea ice concentration and surface heat flux. Arctic sea ice concentration (%) in Jul–Nov from simulations of **a** mid-Holocene climate, **b** pre-industrial climate, and **c** mid-Holocene climate with pre-industrial sea ice concentration (imposed via increased ice albedo). Jul–Nov averaged sea ice concentration differences **d** between the mid-Holocene and pre-industrial simulations, i.e. **a**–**b**, and **e** between the mid-Holocene simulation with pre-industrial sea ice and the pre-industrial simulation, i.e. **c**–**b**. The seasonal cycle of **f** net surface shortwave flux ($F_{SW}\uparrow - F_{SW}\downarrow$) and **g** net surface heat flux ($F_{SW}\uparrow - F_{SW}\downarrow + F_{LW}\uparrow - F_{LW}\downarrow + SHF\uparrow + LHF\uparrow$) averaged in the Arctic Ocean. In **f** and **g**, the red, blue, and black lines denote the mid-Holocene, the mid-Holocene with pre-industrial sea ice, and pre-industrial simulations, respectively

latitude (57°–82°N) zonal-mean westerlies from the lower troposphere to the stratosphere (Fig. 3c). In the upper troposphere and lower stratosphere (around 250–100 hPa), the zonal-mean zonal wind anomalies range from −0.5 to −1.0 m s⁻¹. These westerly wind anomalies are similar to those associated with future projections of Arctic sea ice loss based on RCP 8.5[19,21]. This weakening of the mid-high latitude westerlies induced by sea ice loss is partially offset by insolation forcing in boreal winter, during which the insolation in Northern Hemisphere is weaker than the present by 5–10 W m⁻². Figure 3d shows that the relatively weak insolation forcing in the mid-Holocene strengthens the mid-high latitude westerlies, but this westerly strengthening is confined to the stratosphere, with limited signature in the troposphere. The weakening of mid-high latitude westerlies induced by sea ice loss is accompanied by the weakening of storm

tracks over the North Pacific and North Atlantic Oceans (Fig. 3e). We quantify the storm track intensity using the transient eddy heat flux in the lower troposphere, $\overline{V'T'}$. Here, $()$ and $()'$ denote a time average and a deviation from the time average, respectively, and $<>$ denotes mass-weighted vertical integration from the surface to 700 hPa. This weakening of transient eddy activity is probably because of the reduced baroclinic instability that is dynamically tied to the westerly wind speed[34]. Meanwhile, the effect of the direct insolation forcing on the storm tracks is weaker than that of the sea ice loss: the insolation forcing slightly weakens the zonal-mean westerly winds and thus the transient eddy activity at lower latitudes, around 30°N–50°N (Fig. 3f). These results suggest that mid-Holocene Arctic sea ice loss may have had a stronger influence on wintertime tropospheric westerlies and weather activities than the direct insolation forcing.

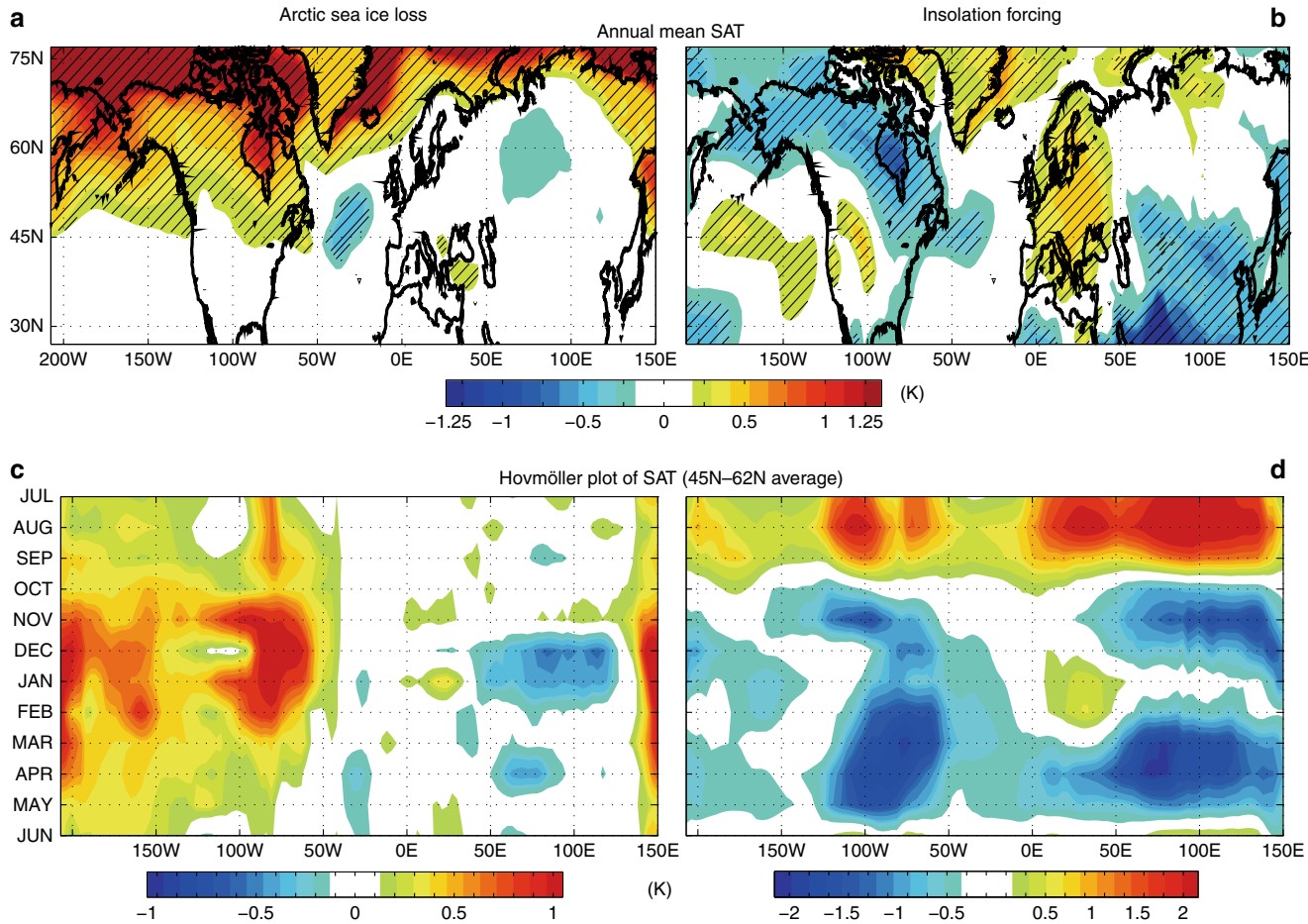

**Fig. 2** Temperature responses to Arctic sea ice loss and insolation. Surface air temperature response to **a**, **c** Arctic sea ice loss and **b**, **d** insolation forcing. **a**, **b** The annual-mean surface air temperature anomalies (K) and **c**, **d** meridionally averaged (45°–62°N), longitude–time Hovmöller plots of anomalous surface air temperature (K). The abscissa longitude and the ordinate is time (months). In **a** and **b**, statistically significant values ($p < 0.05$) are hatched

The Arctic sea ice loss also leads to polar stratospheric warming, especially in late winter (February–March), during which 100-hPa geopotential height increases by 40–50 m (Supplementary Fig. 2). This is consistent with previous studies indicating that sea ice loss induces Rossby waves that penetrate into the stratosphere and weaken the stratospheric polar vortex[16,19,20], especially in late winter[17].

**Sea surface temperature and deep ocean circulations**. These atmospheric circulation and SAT changes are also associated with sea surface temperature (SST) changes. Figure 4c shows that the direct insolation forcing increases the annual-mean North Sea and the Mediterranean Sea SSTs by around 0.2–0.3 K. This SST warming likely contributes to the increase in annual-mean SAT over Europe (Fig. 2b). The Hovmöller plot of the North Atlantic SSTs averaged between 55°W and 10°W indicates that the SST anomalies increase rapidly in summer because of anomalously strong summer insolation, but quickly subside in autumn and become slightly negative in spring (Fig. 4d). In contrast, Arctic sea ice loss produces seasonally persistent SST anomalies in the mid-high latitudes. Arctic sea ice loss increases the annual-mean North Pacific SSTs by around 0.4–0.5 K and produces a localized ~0.5 K decrease in annual-mean SSTs in the central North Atlantic (Fig. 4a). The Hovmöller plot of North Atlantic SSTs shows that the sea ice loss decreases (increases) the mid-latitude (subpolar) North Atlantic SSTs throughout the season (Fig. 4b), and that these anomalies exhibit far less seasonality than those of

the insolation forcing. The summed effects of insolation forcing and sea ice loss exhibit a pattern of North Pacific–subpolar North Atlantic warming and tropical ocean cooling (Supplementary Fig. 4). This is qualitatively consistent with the PMIP3 multi-model average (Supplementary Fig. 4), with the exception of the North Pacific warming.

Via what mechanism does Arctic sea ice loss decrease the North Atlantic SSTs? Consistent with recent modeling studies[35,36], Fig. 5b shows that Arctic sea ice loss weakens the Atlantic meridional overturning circulation (AMOC); the northward mass flux systematically decreases all the way from the subtropics to the subpolar North Atlantic. The decrease in the northward mass flux in the Atlantic Ocean is largest around 35°N–50°N (Fig. 5b) and the poleward margin of this region is where the annual-mean SSTs decrease by around 0.5 K (Fig. 4a, b). However, the AMOC strength slowly recovers to the control simulation (i.e. mid-Holocene climate with preindustrial sea ice) strength after about 180 years (Fig. 5a). After year 200, the AMOC strength anomalies are mostly smaller than 1 Sverdrup (Fig. 5a, c), which is consistent with a previous study[37] showing that the AMOC weakening is a transient response to the Arctic sea ice loss. Therefore, it is unlikely that the AMOC weakening and the associated North Atlantic cooling persisted throughout the mid-Holocene. While the AMOC weakening signal substantially diminishes after year 200, the North Atlantic cooling still persists (Fig. 5g), suggesting that the AMOC weakening is not the only factors causing the North Atlantic

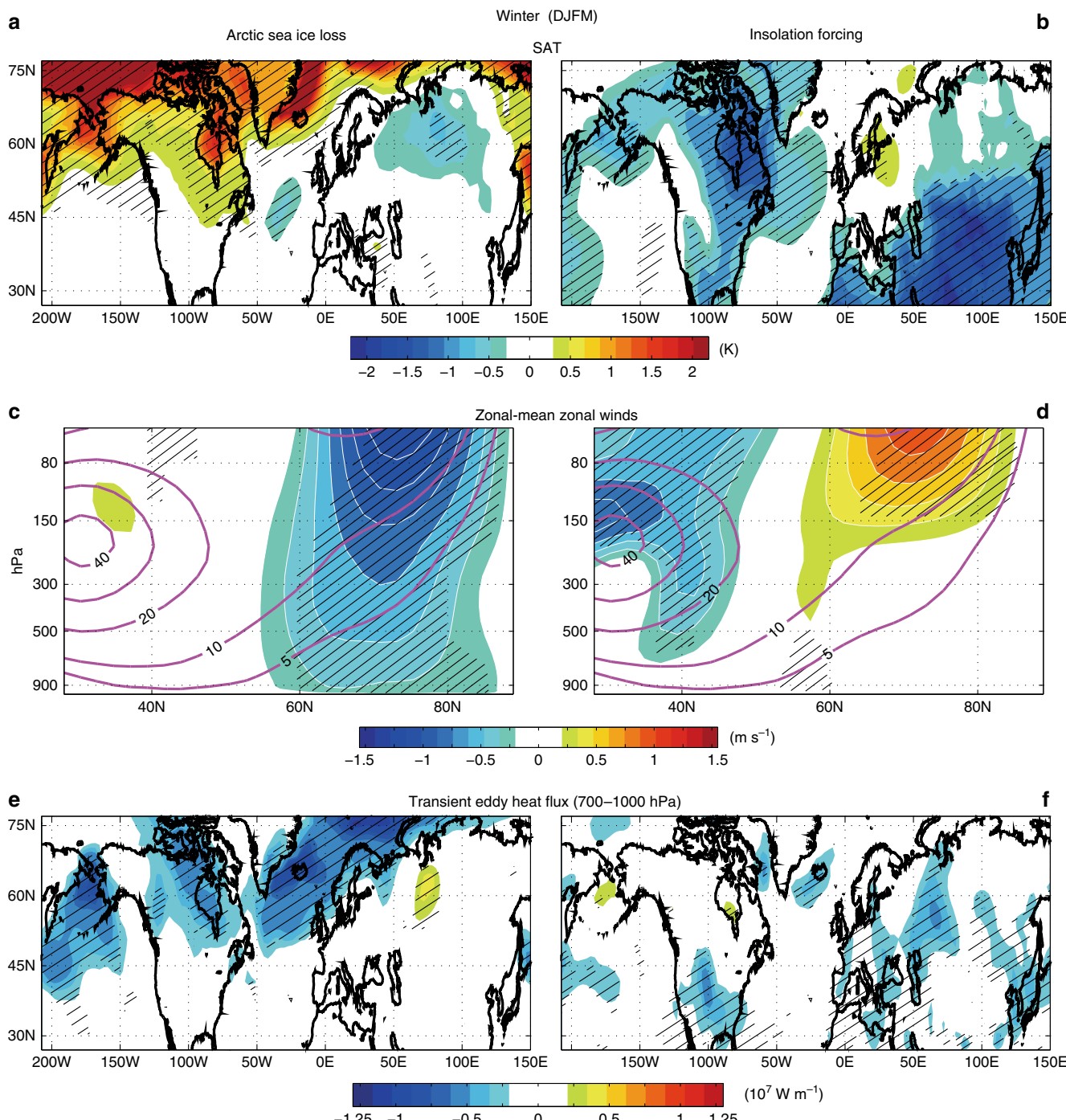

**Fig. 3** Wintertime responses of temperature, westerly winds, and storm tracks. The wintertime (December–March) responses of **a**, **b** surface air temperature (K), **c**, **d** zonal-mean zonal winds (m s$^{-1}$) and **e**, **f** transient eddy heat flux (10$^7$ W m$^{-1}$) to mid-Holocene Arctic sea ice loss (**a**, **c**, **e**) and insolation forcing (**b**, **d**, **f**). Purple lines in **c** and **d** are climatological-mean zonal-mean zonal winds from the mid-Holocene with pre-industrial sea ice simulation. In **e** and **f**, the transient eddy heat flux at each pressure level is integrated from the surface to 700 hPa. For all the plots, statistically significant values ($p < 0.05$) are hatched

cooling. The sea surface height (SSH) anomalies show that Arctic sea ice loss decreases SSH over the mid-North Atlantic (Fig. 5d, e), which weakens the Gulf Stream by reducing the meridional gradient of the SSH. The reduced strength of the Gulf Stream is likely to be a factor for the North Atlantic cooling, which is qualitatively consistent with a previous study[18] showing that the Arctic sea ice loss can change the North Atlantic SST pattern by shifting the Gulf Stream southward.

## Discussion

In summary, our model simulations demonstrate that Arctic sea ice loss driven by anomalously strong summer insolation in mid-Holocene has a profound effect on weather and climate. During mid-Holocene winter, Arctic sea ice loss weakens the mid-high latitudes westerlies, which has been shown to be a robust response to the projected future Arctic sea ice loss across climate models[33]. The consistent zonal-mean westerly response to sea ice loss between the mid-Holocene and the future climate projections

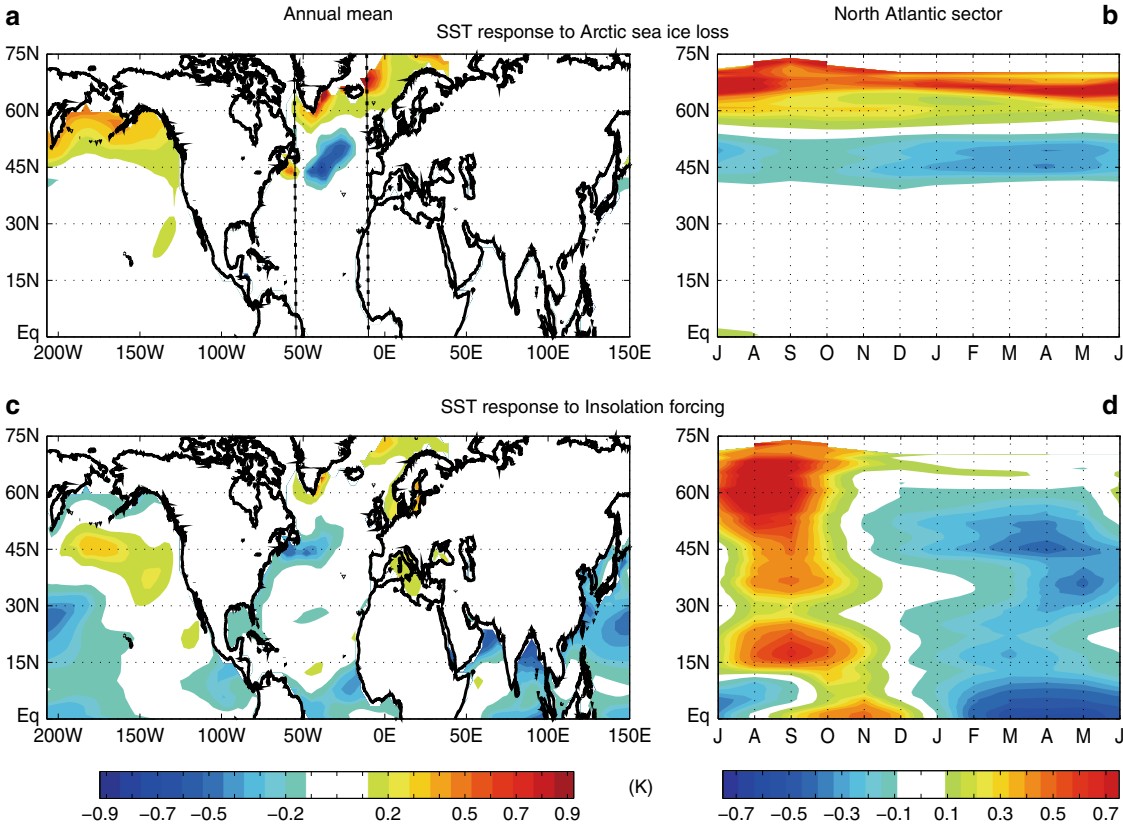

**Fig. 4** Sea surface temperature responses to Arctic sea ice loss and insolation. The SST response to mid-Holocene **a**, **b** Arctic sea ice loss and **c**, **d** insolation forcing. **a**, **c** The annual-mean surface air temperature anomalies (K) and **b**, **d** zonally averaged (55°–10°W; North Atlantic sector), longitude–time Hovmöller plots of anomalous surface air temperature (K). The abscissa is time (months) and the ordinate is latitude. Dotted lines in **a** are longitudinal bounds for the Hovmöller plots

verifies that the impact of Arctic sea ice loss on mid–high latitude climate is relatively insensitive to the mean climate state[38]. Therefore, better understanding of the Holocene climate changes associated with Arctic sea ice may provide guidance on prediction of ongoing and future climate changes in the mid-high latitudes. This negative AO-like zonal-mean response to Arctic sea ice loss in mid-Holocene is accompanied by a zonally asymmetric SAT response in the mid-high latitudes, such as warming over Alaska and Canada and slight cooling over Central and East Asia. The SAT changes induced by sea ice loss are comparable in magnitude to the SAT changes associated with the anomalous winter insolation forcing, which is weaker than the present by 5–10 W m$^{-2}$ in the mid latitudes. In many cases, the circulation and temperature changes driven by Arctic sea ice loss oppose those due to the direct effects of anomalous winter insolation forcing; for example, the sea ice loss-induced weakening of the westerlies is partly offset by the insolation-induced strengthening of westerlies. Additionally, the weaker winter insolation substantially decreases wintertime SAT over North America (Alaska and Canada), whereas Arctic sea ice loss opposes this cooling by increasing SAT over North America by 1–2 K. Our model results suggest that Arctic sea ice cover and direct insolation forcing played distinct roles in mid-Holocene climate change. This separation offers a conceptual framework for interpreting Holocene proxy records of the mid and high latitudes.

## Methods

**Design of climate model experiments.** We perform simulations NCAR CESM1.2.1, a fully coupled model with approximately 1° horizontal resolution (f09g16)[22]. The atmospheric model is the Community Atmospheric Model version

5 (CAM5) with 30 vertical levels, and the ocean component is the Parallel Ocean Program version 2 (POP2) with 60 vertical levels. The land and sea ice components are the Community Land Model version 4 (CLM4) and the Los Alamos sea ice model version 4 (CICE4), respectively.

To distinguish the climatic responses to sea ice loss and anomalous insolation forcing in the mid-Holocene, we perform three different model simulations. First, we perform a 335-year-long Pre-industrial simulation that has constant year 1850 radiative forcing and greenhouse gas concentrations. Second, the mid-Holocene simulation with 6 ka BP insolation and greenhouse gas concentrations is branched off from year 31 of the Pre-industrial simulation and runs for 315 years. Third, another simulation with mid-Holocene forcing is branched off at year 31 of the Pre-industrial run, except the albedo of sea ice is increased globally, and throughout the year, from 0.73 to 0.91 to reflect more sunlight, while the snow albedo over sea ice is not changed. A recent study[21] also used this method (changing sea ice albedo) to isolate the effect of Arctic sea loss on atmospheric circulations.

The three different model simulations can be summarized as:
**0 k**: Pre-industrial control simulation (335-year duration: from year 1 to 335)
**6 ka**: Mid-Holocene climate simulation (315-year duration: from year 31 to 345)
**6 ka with 0 k sea ice**: Mid-Holocene climate with sea ice albedo is increased to 0.91 (316-year duration: from year 31 to 346)

The contributions of Arctic sea ice loss and direct insolation anomalies to mid-Holocene climate, relative to the preindustrial, can be separated as follows:
The contribution of Arctic sea ice loss: (6 ka)–(6 ka with 0 k sea ice)
The contribution of insolation forcing: (6 ka with 0 k sea ice)–(0 k)

In each simulation, we perform analysis using the last 265 years and discard the first 50 years of the simulations 6 ka and 6 ka with 0 k sea ice. In our CESM1 simulations, the Arctic summer sea ice cover quickly responds to the mid-Holocene insolation forcing, within 10 years (Supplementary Fig. 5). The wintertime 200-hPa zonal-mean zonal winds averaged from 65°N to 80°N also responds to the Arctic sea ice loss within few years, although large internal variations of wintertime atmospheric circulations are overlaid the signal (Supplementary Fig. 5). Because of the relatively rapid adjustments of SIC and the associated atmospheric circulations, using the last 150 years of each simulation produces quantitatively similar results. For example, the wintertime SAT and zonal-mean zonal wind anomalies averaged during the last 150 years of model

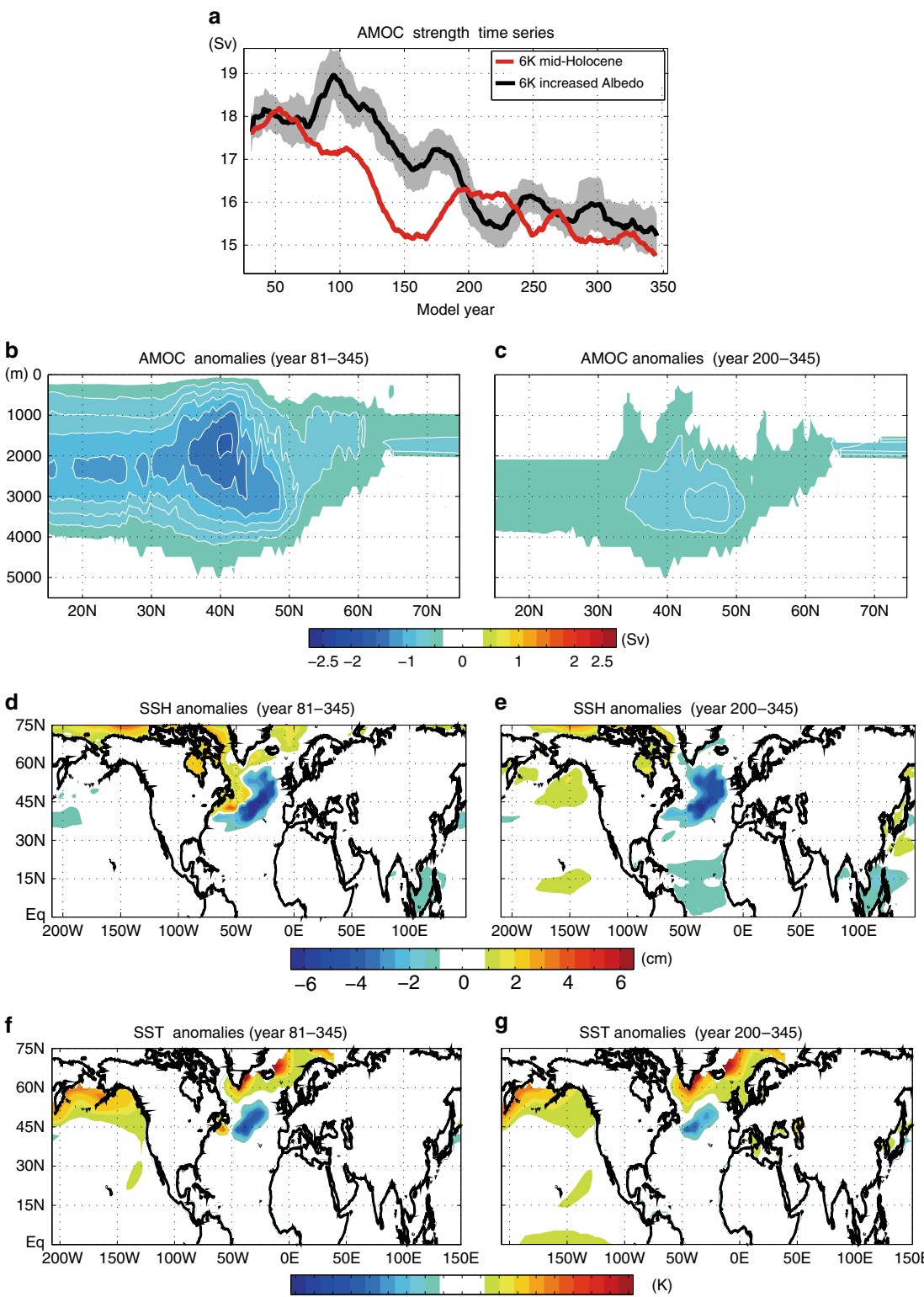

**Fig. 5** The response of Atlantic meridional overturning circulation to Arctic sea ice loss. **a** The 20-year running averages of AMOC strength, defined as the North Atlantic meridional streamfunction (Sv) averaged between depths of 500 and 2500 m, for the mid-Holocene (red) and the mid-Holocene with pre-industrial sea ice (black) simulations. The gray shadings indicate the range of 1 standard deviation from the mean of the mid-Holocene with pre-industrial sea ice (black) simulations. The responses of **b**, **c** AMOC (Sv; $1\,Sv = 10^6\,m^3\,s^{-1}$), **d**, **e** SSH, and **f**, **g** SSTs to Arctic sea ice loss averaged over the model years of **b**, **d**, **f** 81–345 and **c**, **e**, **g** 200–345. Panel **f** is identical to Fig. 4a. In **b** and **c**, the abscissa is latitude and the ordinate is depth (m). In **b** and **c**, only statistically significant values ($p < 0.05$) are shaded

**Table 1 Summary of the PMIP3 (mid-Holocene) and CMIP5 (Pre-Industrial) models**

| Models | Atmos. resolutions | Ocean resolutions | References |
|---|---|---|---|
| BCC-CSM-1 | T42L26 | 360 × 232 L40 | ref. [42] |
| CCSM4 | 0.9° × 1.25° L26 | 320 × 384 L60 | ref. [43] |
| CNRM-CM5 | T127L31 | 362 × 292 L42 | ref. [44] |
| CSIRO-Mk3-6-0 | T63L18 | 192 × 192 L31 | ref. [45] |
| FGOALS-g2 | 2.81° × 2.81° L26 | 360 × 196 L30 | ref. [46] |
| GISS-E2-R | 2.0° × 2.5° L40 | 288 × 180 L32 | ref. [47] |
| IPSL-CM5A-LR | 1.875° × 3.75° L39 | 182 × 149 L31 | ref. [48] |
| MIROC-ESM | 2.8° × 2.8° L80 | 256 × 192 L44 | ref. [49] |
| MPI-ESM-P | T63L47 | 256 × 220 L40 | ref. [50] |
| MRI-CGCM3 | TL159L48 | 364 × 368 L51 | ref. [51] |

simulations are very similar to those averaged during the last 265 years (compare Supplementary Fig. 5 and Fig. 3). These results are generally consistent with previous studies showing that the adjustment timescales of atmosphere and the ocean mixed layer to a radiative forcing given as a step function are only a few years[39,40]. Discarding the first 50 years of model integration, therefore, captures the equilibrium response of atmosphere, whereas the deep ocean circulation takes several hundred years to reach an equilibrium[37,39,41] (Fig. 5a).

**Validation of climate model outputs**. The increased ice albedo in the 6 ka with 0 k sea ice experiment prevents Arctic sea ice loss in summer–early autumn in response to the anomalously strong mid-Holocene insolation. However, this increase in ice albedo has little influence on Southern Hemisphere sea ice (see Supplementary Fig. 6), which is mostly covered by much thicker snow cover than Arctic sea ice. The increased ice albedo simulation maintains the Arctic sea ice cover by reflecting anomalously strong 6 ka summer insolation, keeping SIC anomalies within 5% of the preindustrial simulation in summer and autumn (Fig. 1e). We find that sea ice albedos ranging from 0.90 to 0.92 keep the Arctic sea ice cover almost unchanged; the differences of Arctic SIC between this increased mid-Holocene ice albedo and the pre-industrial control simulations are within 5%. Increasing the ice albedo above 0.93 increases the annual-mean SIC and sea ice thickness by more than 5% relative to the pre-industrial climate. The Arctic sea ice cover is dynamically tied to net surface heat flux ($F_{net}\uparrow$), especially net surface shortwave flux ($F_{SW}\uparrow - F_{SW}\downarrow$): $F_{net}\uparrow = F_{SW}\uparrow - F_{SW}\downarrow + F_{LW}\uparrow - F_{LW}\downarrow + SHF\uparrow + LHF\uparrow$, where $F_{SW}\uparrow$ and $F_{SW}\downarrow$ are upward and downward shortwave radiative fluxes, respectively. $F_{LW}\uparrow$ and $F_{LW}\downarrow$ are upward and downward longwave radiative fluxes, respectively, and $SHF$ and $LHF\uparrow$ denote sensible and latent heat fluxes, respectively. The net surface shortwave flux simulated by the increased mid-Holocene ice albedo experiment is almost identical to that of the pre-industrial control simulation (Fig. 1f), verifying the close relationship between summer sea ice cover and net surface shortwave flux. Similarly, the net surface heat flux simulated by the increased mid-Holocene ice albedo experiment is almost identical to that of the pre-industrial control simulation (Fig. 1g); net surface heat fluxes between these two simulations are very similar not only in summer but also in autumn and winter, during which the net surface heat flux is upward mainly due to large sensible and latent heat fluxes ($SHF\uparrow + LHF\uparrow$).

To assess whether these model simulation produced different climate conditions, a statistical significance test was performed for seasonally averaged variables. Specifically, a two-sample $t$-test was performed for each model grid point to determine whether the interannual variations of two different simulations (265 time-series data for each simulation) had equal means or not. These statistically significant values are presented as 'hatches' in the figures.

**Comparison with other climate model simulations**. To assess the realism of mid-Holocene climate simulated by CESM1, we also examine ten coupled GCMs from the PMIP3 and CMIP5. We evaluated the SAT and SST fields by examining the differences between PMIP3 mid-Holocene and CMIP5 Pre-Industrial simulations. A list of the PMIP3–CMIP5 models and their Atmosphere and Ocean resolutions are provided in Table 1.

**Code availability**. Code from this study (mostly MATLAB scripts) is available from the corresponding author upon request.

## Data availability

Model (NCAR CESM1.2.1) output from this study is available on Earth Linux cluster sever at Korea Institute of Geoscience and Mineral Resources (KIGAM), where all the monthly output variables are stored. Several daily output variables are also available. These monthly and daily data are available from the corresponding author upon reasonable request.

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

## Acknowledgements

H.-S.P. was supported by the Basic Research Project (GP2017-013) of the Korea Institute of Geoscience and Mineral Resource (KIGAM), Ministry of Science, ICT, and Future Planning. S.-J.K. was supported by KOPRI project no. PE18130. K.-H.S. and S.-W.S. were supported by the National Research Foundation of Korea (NRF) grants NRF-2018R1A2A2A05018426 and NRF-2018R1A5A1024958, respectively. A.L.S. was supported by the National Science Foundation under grant no. ANT-1543388.

## Author contributions

H.-S.P. initiated the project and carried out the analysis under the guidance of S.-J.K. and A.L.S. The manuscript was initially written by H.-S.P. and was edited by A.L.S. K.-H.S. and S.-W.S. were in charge of the discussion of the wintertime atmospheric circulations (Fig. 3). The PMIP3 data were collected and re-gridded by S.-Y.K. All authors contributed to the interpretations of the results and the discussion of the manuscript.

## Additional information

**Competing interests:** The authors declare no competing interests.

