## [Peer Review File · Nature Communications]

Reviewers' Comments:

Reviewer #1:

Remarks to the Author:

The manuscript entitled "Desentangling the impacts of Arctic sea ice loss and insolation forcing on mid-Holocene climate" has used climate modelling by separating the impact of sea-ice loss and insolation forcing on mid-Holocene northern mid-high latitudes climate. They conclude that the sea-ice loss plays an important role on mid-Holocene climate change. The modelling work and results interpretation are following the similar ideas of Blackport and Kushner (2017) by applying to mid-Holocene climate. In this regard this work is not novel. The major results in this work largely depend on the modelling methods, which I found is not convincing.

Exp3 is designed by increasing the sea-ice albedo to 0.91, in order to keep a fixed PI sea-ice coverage as shown in Fig1. It does not mention if this fixed albedo 0.91 is applied to everywhere and each month. The change in albedo will result in changes in absorbed shortwave radiation at surface where there is no snow, since they did not change the snow albedo, and in this case the surface energy can be largely different from the real PI sea-ice case, even though Exp1 and Exp3 have the same SIC. Because the albedo is fixed in EXP3, the sea-ice will not melt with whatever insolation, the sea-ice albedo feedback during summer does not work. Therefore, a linear difference between exp2 and exp3 does not result in a reasonable sea-ice loss from PI to MH, because the surface energy over sea-ice is largely distorted, especially in summer. The linear difference between exp1 and exp3 does not sort out a pure insolation effect either, because over the same SIC there are different surface energy due to the different sea-ice albedo.

With this regard on the major method for this work is unsound, I don't think the work is suitable to publish in Nature communication.

Reviewer #2:

Remarks to the Author:

1. In this contribution, the authors seek to separate the roles of sea ice loss and insolation forcing in driving climate anomalies in the mid-Holocene. They find that sea ice loss signal drives significant changes in surface temperature and aspects of the circulation. The two effects sometime reinforce and sometime cancel the signal.

2. This is a clearly written and well motivated study whose topic might potentially be of interest to a broad readership. However, the method has a weakness that needs to be addressed before I could recommend the study for publication. Namely, the length of the simulations used in the study - a 50 year control simulation, branching to 100 year perturbation simulations of which the entire 100 years (or just the last 50 years) are used, appears to me to be too short to reliably represent the equilibrated adjustment to the forcing applied.

3. This is not a mere technicality: Deser et al. (2015, footnote 18) use over 300 years to simulate the response to sea ice loss. Blackport and Kushner (2016, not cited in this paper), used an identical methodology to Blackport and Kushner (2017), but with CCSM4. They found that the AMOC response of the kind reported here was a transient signal that disappeared after about three hundred years of simulation (Figure 1 of that paper). The transient phase of the CCSM4 simulation, which Blackport and Kushner sampled over the first 50 years, was characterized by non-robust variability over Eurasia and in the stratosphere. Several realizations were required to adequately sample the transient phase.

It is reassuring that the total response to mid-Holocene forcing appears to be robust across CESM1 and the other PMIP3 simulations (as shown in the supplementary figures). The main issue, however, is that separating the response to ice loss in isolation, especially since it often opposes

the response to solar forcing, requires good sampling. I'm not convinced that the authors have achieved this.

4. Given this, I recommend as follows:

a. Extending the integrations in the current study to guarantee the robustness of the results. This would reduce the risk that the responses found here are strongly contaminated by internal atmosphere-ocean coupled variability. I suggest that at least 200 years are required to obtain a robust response for polar regions and for the Eurasian/Pacific sector. 300 years is a better standard found in several other publications.

b. Clarifying whether the authors are interested in the transient adjustment or the equilibrated response (I would expect the latter would be of more interest).

c. Whatever approach is taken, the authors need to provide graphical evidence through time series that whatever responses obtained are robust to transience and to internal variability. E.g. they should show a time series of the AMOC index response to provide the reader with a sense of how robustly sampled the signal is.

5. The authors should clarify that sea ice loss is a conceptually distinctive driver from insolation. I know they understand that the sea ice field (concentration, thickness, distribution) consists of state variables internal to the system, while insolation is an external boundary forcing. The authors have done a good job calibrating the ice albedo to mimic the coupled model response under mid-holocene forcing. Readers might be confused about this suggesting that sea ice an external forcing rather than a part of the coupled system responding to the external forcing.

6. The authors should cite the Blackport and Kushner (2016) study and the recent review of sea ice loss in coupled simulations by Screen et al. (2018).

7. Note the typos on lines 62, 117 and 139, also that reference 33 has the wrong title.

References:

Blackport, R. and P.J. Kushner, 2016: The Transient and Equilibrium Climate Response to Rapid Summertime Sea Ice Loss in CCSM4. *J. Climate*, 29, 401–417, <https://doi.org/10.1175/JCLI-D-15-0284.1>

Screen, J.A., C. Deser, D.M. Smith, X. Zhang, R. Blackport, P.J. Kushner, T. Oudar, K.E. McCusker & L. Sun, 2018: Consistency and discrepancy in the atmospheric response to Arctic sea ice loss across climate models, *Nature Geosci.* <https://www.nature.com/articles/s41561-018-0059-y>

Reviewer #3:

Remarks to the Author:

See attached file.

Review of NCOMMS-18-03003-T

Disentangling the impacts of Arctic sea ice loss and insolation forcing on mid-Holocene climate
by H.-S. Park, S.-J. Kim, K.-H. Seo, A.L. Stewart, and S.-Y. Kim

Overview

This study is a well written, straight-forward analysis of model simulations of the Holocene epoch, when solar insolation exhibited a larger seasonal cycle and Arctic sea ice was reduced. They use forcing for this time period to model the separate roles of insolation variation and sea-ice loss in contributing to the response of the Northern Hemisphere atmospheric circulation. Model simulations suggest that insolation changes alone would have cooled both major Northern Hemisphere continents, but the loss of sea ice counteracts the cooling in North America and exacerbates the cooling over Asia. Sea-ice loss also warms the polar stratosphere and weakens the upper-level westerly flow, especially in winter.

I have no major criticisms of this study, and I recommend it be published once the comments below are addressed. I believe it would be of great interest to the readership of *Nature Communications*, and would make an important contribution to the rapidly progressing research on this topic.

Specific comments

1. My main suggestion is to add discussion at the end of the paper to put these findings into context with present-day conditions. The abstract states that this study is motivated by “...recent studies identifying Arctic sea ice loss as one of the main drivers of future climate change...” so I recommend that the authors circle back to this motivation and connect their results with today’s changing climate and other recent studies.
2. Line 107-108: Does the model simulate a realistic seasonal cycle in North Atlantic SSTs? In today’s climate, the range is about 7°C. Does this range increase under Holocene forcing?
3. 142-145: To get zonal wind anomalies with height shown in Fig. 5e,f, height changes must occur not only in the stratosphere, but throughout the atmospheric column, increasing with height. It would be illuminating to compare these zonal wind responses to those from RCP 8.5 projections of the future, in which solar forcing is not changed appreciably but sea ice loss is similar to Holocene conditions.
4. 148-151: Authors describe how planetary wave heat fluxes are affected, what about those associated with transient eddies? Does the finding that sea-ice-induced increases in stationary eddy fluxes are comparable to the increases in the total flux imply that low wavenumber circulation regimes are favored in low sea-ice conditions? If so, this would support other studies suggesting that reduced sea ice contributes to higher amplitude jet-stream configurations. A bit more explanation of the atmospheric changes and their impacts beyond only meridional fluxes and surface temperatures (e.g., storm tracks, precipitation patterns, long-duration conditions, etc.) would be helpful.

5. Figure 3: Please mark longitude bounds on map (3a) that correspond to the Hovmöller plots in b and d.
6. Figure 4 caption: State flux sign convention (I assume upward is positive).
7. Figure 5 caption: Identify purple contours in e and f.

Response to Reviewer #1:

We thank Reviewer 1 for thoughtful comments. In order to address the reviewer's central concern that the surface energy balance may be unphysically modified by increasing the ice albedo, we added a plot showing the seasonal cycle of net surface heat flux simulated by the three individual experiments (Figs. 1f, 1g). The net surface shortwave flux (Fig. 1f) and net surface heat flux (Fig. 1g) simulated by the increased mid-Holocene ice albedo experiment are almost identical to those of the pre-industrial control simulation, verifying the close relationship between summer sea ice cover and net surface shortwave flux.

Specific comments:

Reviewer: The manuscript entitled “Disentangling the impacts of Arctic sea ice loss and insolation forcing on mid-Holocene climate” has used climate modelling by separating the impact of sea-ice loss and insolation forcing on mid-Holocene northern mid-high latitudes climate. They conclude that the sea-ice loss plays an important role on mid-Holocene climate change. The modelling work and results interpretation are following the similar ideas of Blackport and Kushner (2017) by applying to mid-Holocene climate. In this regard this work is not novel.

Response: We agree with the reviewer that our results are qualitatively consistent with the findings of Blackport and Kushner (2017), who focused on separating the impacts of projected future sea ice loss from future greenhouse radiative forcing. In this study we translate a key idea from their study, specifically that radiative forcing perturbations can be distinguished from perturbations due to sea ice retreat, to the mid-Holocene climate, where such a distinction between forcing perturbations has not previously been made. We did not intend to give the impression that the methodology for separating these forcing perturbations was novel, and in our revised manuscript we have reviewed the Methods section to ensure that this is conveyed clearly.

Reviewer: The major results in this work largely depend on the modelling methods, which I found is not convincing. Exp3 is designed by increasing the sea-ice albedo to 0.91, in order to keep a fixed PI sea-ice coverage as shown in Fig1. It does not mention if this fixed albedo 0.91

is applied to everywhere and each month. The change in albedo will results in changes in absorbed shortwave radiation at surface where there is no snow, since they did not change the snow albedo, and in this case the surface energy can be largely different from the real PI sea-ice case, even though Exp1 and Exp3 have the same SIC.

Response: We agree with the reviewer that it is important to preserve the surface energy budget as closely as possible in order to produce a controlled set of experiments. We have now provided additional figure panels to show the net surface shortwave flux (Fig. 1f) and net surface heat flux (Fig. 1g) simulated by the increased mid-Holocene ice albedo experiment. These fluxes are almost identical to those of the pre-industrial control simulation. We would not expect increasing the snow albedo to substantially alter our results because the snow albedo is already very high in CESM1-CAM5 (visible snow albedo = 0.98) and the anomalously strong mid-Holocene insolation was primarily confined to the summer months, during which the simulated snow concentration in the Arctic is low (please see a Figure below). We also have now clarified that the ice albedo increase is applied globally and throughout the year in the Methods section.

Figure for reviewer 1: CESM1-simulated Jun-Jul-Aug mean snow fraction (%) during the mid-Holocene. The snow fraction is mostly below 25% in summer, except the multiyear ice pack (eastern Canada – northern Greenland sector), where snow fraction is around 30– 40%.

Reviewer: Because the albedo is fixed in EXP3, the sea-ice will not melt with whatever insolation, the sea-ice albedo feedback during summer does not work. Therefore, a linear difference between exp2 and exp3 does not result in a reasonable sea-ice loss from PI to MH, because the surface energy over sea-ice is largely distorted, especially in summer. The linear difference between exp1 and exp3 does not sort out a pure insolation effect either, because over the same SIC there are different surface energy due to the different sea-ice albedo.

Response: Even with increased sea ice albedo, strong summer insolation can completely melt the sea ice. The sea ice albedo feedback is still active because there is still a large contrast between the albedo of the ice (0.91) and the open water (0.06). As mentioned above, the net surface shortwave flux (now shown in the new Fig. 1f) simulated by the increased mid-Holocene ice albedo experiment is almost identical to that of the pre-industrial control simulation, verifying that our experiments accurately control for the surface energy budget, and thereby the difference between experiments 1 and 3 isolates the effect of insolation. The sea ice albedo in experiment 3 has been chosen specifically to reproduce the Arctic sea ice extent in the PI to within 5% difference, again while preserving the surface energy budget, and so isolates the impact of sea ice loss.

Response to Reviewer #2:

We thank Reviewer 2 for helpful and constructive comments. Following the reviewer's suggestion, we substantially increased the length of the simulations from 150 years to 315–335 years. The increased length of simulations might better represent the equilibrated adjustment to the sea ice loss as well as to the direct insolation forcing. By doing so, we were able to clarify whether the AMOC weakening and the North Atlantic cooling driven by sea ice loss are transient responses or not.

Specific comments:

Reviewer: It is reassuring that the total response to mid-Holocene forcing appears to be robust across CESM1 and the other PMIP3 simulations (as shown in the supplementary figures). The main issue, however, is that separating the response to ice loss in isolation, especially since it often opposes the response to solar forcing, requires good sampling. I'm not convinced that the authors have achieved this. Given this, I recommend as follows:

a. Extending the integrations in the current study to guarantee the robustness of the results. This would reduce the risk that the responses found here are strongly contaminated by internal atmosphere-ocean coupled variability. I suggest that at least 200 years are required to obtain a robust response for polar regions and for the Eurasian/Pacific sector. 300 years is a better standard found in several other publications.

Response: Following the reviewer's suggestion, we extended the integrations of each simulation. The pre-industrial control simulation has been extended to 335 years. The mid-Holocene (6Ka BP) and the mid-Holocene with increased albedo simulations have been extended to 315 and 316 years, respectively.

b. Clarifying whether the authors are interested in the transient adjustment or the equilibrated response (I would expect the latter would be of more interest).

Response: Following the reviewer's suggestion, we clarified in the 'Methods' section that we focus on the equilibrated response rather than the transient adjustment, which usually appears in the first ~50 years of the applied forcing. This is briefly discussed in lines 222 – 224.

c. Whatever approach is taken, the authors need to provide graphical evidence through time series that whatever responses obtained are robust to transience and to internal variability. E.g. they should show a time series of the AMOC index response to provide the reader with a sense of how robustly sampled the signal is.

Response: Following the reviewer's suggestion, we provide graphical evidence in Figure 5 that the AMOC weakening is a transient response, but the North Atlantic cooling is not necessarily a transient response to the sea ice loss, and is associated with persistent SSH anomalies over the North Atlantic (Lines 157 – 175).

Reviewer: The authors should clarify that sea ice loss is a conceptually distinctive driver from insolation. I know they understand that the sea ice field (concentration, thickness, distribution) consists of state variables internal to the system, while insolation is an external boundary forcing. The authors have done a good job calibrating the ice albedo to mimic the coupled model response under mid-holocene forcing. Readers might be confused about this suggesting that sea ice an external forcing rather than a part of the coupled system responding to the external forcing.

Response: In this revised version, we used a terminology of 'direct insolation forcing' instead of 'insolation forcing' to clarify that Arctic sea ice loss is a conceptually distinct driver from insolation. To clarify that the Arctic sea ice loss is part of the coupled system, we added a plot showing the seasonal cycle of net surface heat flux simulated by the three individual experiments (Figs. 1f, 1g). The net surface shortwave flux (Fig. 1f) and net surface heat flux (Fig. 1g) simulated by the increased mid-Holocene ice albedo experiment are almost identical to those of the pre-industrial control simulation, verifying the close relationship between summer sea ice cover and net surface shortwave flux.

Reviewer: The authors should cite the Blackport and Kushner (2016) study and the recent review of sea ice loss in coupled simulations by Screen et al. (2018).

Response: In this revised version, we cited Blackport and Kushner (2016) to clarify the

differences between the transient and equilibrium responses to Arctic sea ice loss. We also cited Screen et al. 2018 (Line 110).

Reviewer: Note the typos on lines 62, 117 and 139, also that reference 33 has the wrong title.

Response: These typos are corrected. Thank you for pointing out these mistakes.

Response to Reviewer #3:

We thank Reviewer 3 for encouraging words and helpful comments. Following the reviewer's suggestion, we added a plot showing the response of transient eddies to the sea ice loss (Fig. 3e). We also highlighted the different atmospheric responses to sea ice loss between early winter (Dec-Jan) and late winter (Feb-Mar) in Supplementary Fig. 2, showing that the polar stratospheric warming signal is far stronger in the late winter than in the early winter. Finally, we substantially increased the length of the simulations from 150 years to 315–335 years to better represent the equilibrated adjustment to the sea ice loss.

Specific comments:

Reviewer comments (1): My main suggestion is to add discussion at the end of the paper to put these findings into context with present-day conditions. The abstract states that this study is motivated by “...recent studies identifying Arctic sea ice loss as one of the main drivers of future climate change...” so I recommend that the authors circle back to this motivation and connect their results with today's changing climate and other recent studies.

Response: Following the reviewer's suggestion, we briefly discussed how our findings regarding the mid-Holocene climate relate to the impacts of ongoing and future sea ice loss.

Lines 180 – 185: “The consistent zonal-mean westerly response to sea ice loss between the mid-Holocene and the future climate projections verifies that the impact of Arctic sea ice loss on mid–high latitude climate is relatively insensitive to the mean climate state³⁸. Therefore, better understanding of the Holocene climate changes associated with Arctic sea ice may provide guidance on prediction of ongoing and future climate changes in the mid-high latitudes”.

Reviewer comments (2): Line 107-108: Does the model simulate a realistic seasonal cycle in North Atlantic SSTs? In today's climate, the range is about 7°C. Does this range increase under Holocene forcing?

Response: As the reviewer pointed out, the seasonal range of North Atlantic SSTs averaged

from 35° N to 60° N is **7.07° C** based on observations (please see a Figure below). The seasonal range simulated by CESM1-CAM5 pre-industrial forcing is **6.54° C**, which underestimates that of the present observation by 7%. CESM1-CAM5 also shows a slight warm bias. As the reviewer pointed out, the mid-Holocene insolation forcing substantially increases the seasonal range: the seasonal range of the North Atlantic SSTs under mid-Holocene insolation forcing is **7.36° C**, which is about 13% larger than that of the pre-industrial climate.

Figure for reviewer 3: The seasonal cycle of mean North Atlantic SST from the HadISST (black line), CESM1-CAM5 simulations under 6K mid-Holocene forcing (red line) and under pre-industrial forcing (blue line).

Reviewer comments (3): 142-145: To get zonal wind anomalies with height shown in Fig. 5e,f, height changes must occur not only in the stratosphere, but throughout the atmospheric column, increasing with height. It would be illuminating to compare these zonal wind responses to those from RCP 8.5 projections of the future, in which solar forcing is not changed appreciably but sea ice loss is similar to Holocene conditions.

Response: In this revised version, we deleted sentences arguing the direct causal relationship between the polar stratospheric warming and the negative AO-like westerly weakening. We also presented in Supplementary Fig. 2 that the polar stratospheric warming signal is far

stronger in late winter (Feb-Mar) than in early winter (Dec-Jan), which is consistent with a previous study. Finally, following the reviewer's suggestion, we clarified that our zonal-mean zonal wind response to sea ice loss is very similar to that associated with RCP 8.5 projected future sea ice loss. The related sentences have been added on lines 114–117: “In the upper troposphere and lower stratosphere (around 250–100 hPa), the zonal-mean zonal wind anomalies range from -0.5 to -1.0 m s⁻¹. These westerly wind anomalies are similar to those associated with future projections of Arctic sea ice loss based on representative concentration pathway 8.5 (RCP8.5)^{19, 21}”.

Reviewer comments (4): 148-151: Authors describe how planetary wave heat fluxes are affected, what about those associated with transient eddies? Does the finding that sea-ice-induced increases in stationary eddy fluxes are comparable to the increases in the total flux imply that low wavenumber circulation regimes are favored in low sea-ice conditions? If so, this would support other studies suggesting that reduced sea ice contributes to higher amplitude Jetstream configurations. A bit more explanation of the atmospheric changes and their impacts beyond only meridional fluxes and surface temperatures (e.g., storm tracks, precipitation patterns, long-duration conditions, etc.) would be helpful.

Response: Following the reviewer's suggestion, we added a plot showing the response of transient eddies to sea ice loss (Fig. 3e) and to the direct effect of insolation forcing (Fig. 3f) (Please see lines 122–131). Figure 3e shows that the weakening of mid-high latitude westerlies induced by sea ice loss is accompanied by weakening of the storm tracks over the North Pacific and North Atlantic Oceans.

Reviewer comments (5): Figure 3: Please mark longitude bounds on map (3a) that correspond to the Hovmöller plots in b and d.

Response: (Figure 3 is now Figure 4). To make the plots visually clear to readers, we adjusted the latitudinal ranges between the left column (Figs 4a, c) and the right column (Fig. 4b, d) be identical. We also marked the longitudinal bounds in Fig. 4a (black-dotted line).

Reviewer comments (6): Figure 4 caption: State flux sign convention (I assume upward is positive).

Response: In this revised version, we placed less emphasis on the AMOC weakening, which our extended simulations revealed to be a transient response to the sea ice loss. We deleted the plot showing the surface heat flux anomalies over the North Atlantic & Arctic Oceans.

Reviewer comments (7): Figure 5 caption: Identify purple contours in e and f.

Response: (Figure 5 is now Figure 3). Thank you for pointing this out. We illustrated the purple contours in the caption of Figure 3: “Purple lines in (c, d) are climatological-mean zonal-mean zonal winds from the mid-Holocene with pre-industrial sea ice simulation”.

Reviewers' Comments:

Reviewer #1:

Remarks to the Author:

I am happy with the clarification regarding my questions on surface energy balance. The manuscript is now presented in a clear way and can consider for publication. I have two minor comments for a further improvement.

1. The effect separation methods is the base for this work, therefore I suggest that the author briefly introduce how do they design the sensitivity experiments in earlier part of the manuscript. It would be helpful readers if authors can add a few lines to describe the method at line 51-52, instead saying "described in the Methods".

2. The authors have responded to the reviewer#2' comments on equilibrium response, and added a AMOC evolution in Fig5a. The time series show that apparently both runs do not reach a equilibrium state until after model year 200, it looks there is still a decreasing trend even after 200 years, especially for the 6k experiment (red line). I encourage for another 100 years simulation to make sure an equilibrium or quasi-equilibrium state is reached. The authors mentioned in the analyses have used last 265 years, to me this time window still contains the transient part. And they mentioned that "using the last 150 years produces similar results", in this case I think they should use last 150 years which more represent an equilibrium state. And it would be good to show AMOC from all three simulations in fig5a.

Reviewer #2:

Remarks to the Author:

I recommend acceptance of this ms.

The reviewers have responded to my comments (I was Reviewer 2 for the original submission), and I commend them in particular for obtaining the computational resources required to extend the simulations. This was an important step to providing confidence in the results.

It is useful to see the AMOC time series in Figure 5, but it does point out a potential minor problem in the ms: the time series it does not support the statement that transients are usually only present in the first 50 years (l.224), since the AMOC time series take over 150 years to settle down. I don't think the cited reference (Blackport and Kushner) claim that 50 years is an adjustment timescale, although the first fifty years is the period they use to characterize the transient response. It is reassuring that the last 150 years of simulation produces similar results (l.225). I suggest clarifying this point.

Response to Reviewer #1:

Reviewer: (1) The effect separation methods is the base for this work, therefore I suggest that the author briefly introduce how do they design the sensitivity experiments in earlier part of the manuscript. It would be helpful readers if authors can add a few lines to describe the method at line 51-52, instead saying “described in the Methods”.

Response: We agree with the reviewer that briefly introducing the key method in the ‘introduction’ section can be helpful for readers, especially those who are interested in the methodology. We have now rewritten this sentence and the following one as: “We perform three simulations to separate the effect of sea ice loss from that of direct insolation: one with pre-Industrial insolation and sea ice, one with mid-Holocene insolation and sea ice, and one with mid-Holocene insolation and pre-Industrial sea ice (see Methods).”

Reviewer: (2) The authors have responded to the reviewer#2' comments on equilibrium response, and added a AMOC evolution in Fig5a. The time series show that apparently both runs do not reach a equilibrium state until after model year 200, it looks there is still a decreasing trend even after 200 years, especially for the 6k experiment (red line). I encourage for another 100 years simulation to make sure an equilibrium or quasi-equilibrium state is reached. The authors mentioned in the analyses have used last 265 years, to me this time window still contains the transient part. And they mentioned that "using the last 150 years produces similar results”, in this case I think they should use last 150 years which more represent an equilibrium state.

Response: As the reviewer pointed out, the deep ocean circulation such as AMOC takes several hundred years to reach an equilibrium. In order to address the reviewer’s concern, we added plots (Supplementary Fig. 5) showing that the Arctic summer sea ice cover quickly responds to the mid-Holocene insolation forcing in our CESM1 simulations. The wintertime zonal-mean zonal winds also respond to the Arctic sea ice loss within few years. Therefore, the wintertime temperature and zonal-mean zonal wind anomalies averaged during the last 150 years of model simulations are very similar to those averaged during the last 265 years (compare Supplementary Fig. 5 and Fig. 3). These results are consistent with previous studies

(Olivie et al. 2012; Rugenstein et al. 2016) showing that the adjustment timescales of atmosphere and the ocean mixed layer to a radiative forcing given as a step function are only a few years. Therefore, discarding the first 50 years of model integration captures the equilibrium response of atmosphere.

We have now added the following text to the Methods section to clarify these points: “In our CESM1 simulations, the Arctic summer sea ice cover quickly responds to the mid-Holocene insolation forcing, within 10 years (Supplementary Fig. 5). The wintertime 200-hPa zonal-mean zonal winds averaged from 65°N to 80°N also responds to the Arctic sea ice loss within few years, although large internal variations of wintertime atmospheric circulations are overlaid the signal (Supplementary Fig. 5). Because of the relatively rapid adjustments of SIC and the associated atmospheric circulations, using the last 150 years of each simulation produces quantitatively similar results. For example, the wintertime SAT and zonal-mean zonal wind anomalies averaged during the last 150 years of model simulations are very similar to those averaged during the last 265 years (compare Supplementary Fig. 5 and Fig. 3). These results are generally consistent with previous studies showing that the adjustment timescales of atmosphere and the ocean mixed layer to a radiative forcing given as a step function are only a few years^{39, 40}. Discarding the first 50 years of model integration therefore captures the equilibrium response of atmosphere, whereas the deep ocean circulation takes several hundred years to reach an equilibrium^{37, 39, 41} (Fig. 5a).”

Regarding the SSTs: we have an existing figure showing that the SST anomalies averaged over the last 150 years (Fig. 5g) are qualitatively similar to those averaged over the last 265 years (Fig. 5f).

Reviewer: And it would be good to show AMOC from all three simulations in fig5a.

The figure below shows the AMOC strength time series from all three simulations:

<Fig A: Same as Fig. 5a, except that the AMOC strength time series from the pre-industrial control simulation (blue line) is added>

This plot shows that the AMOC strength of pre-industrial control simulation (blue line) generally follows that of 6 ka increased albedo simulation (black line), verifying that the AMOC strength is largely controlled by Arctic sea ice loss rather than the direct insolation forcing. However, because Figure 5 is focusing on the AMOC and SST changes driven by **Arctic sea ice loss**, we are concerned that adding this blue line (the AMOC strength from the pre-industrial control simulation) can be confusing to readers. We have therefore left the format of Fig. 5a unchanged.

Response to Reviewer #2:

Reviewer: It is useful to see the AMOC time series in Figure 5, but it does point out a potential minor problem in the ms: the time series it does not support the statement that transients are usually only present in the first 50 years (1.224), since the AMOC time series take over 150 years to settle down. I don't think the cited reference (Blackport and Kushner) claim that 50 years is an adjustment timescale, although the first fifty years is the period they use to characterize the transient response. It is reassuring that the last 150 years of simulation produces similar results (1.225). I suggest clarifying this point.

Response: In order to address the reviewer's concern, we added plots (Supplementary Fig. 5) showing that the Arctic summer sea ice cover quickly responds to the mid-Holocene insolation forcing in our CESM1 simulations. The wintertime zonal-mean zonal winds also respond to the Arctic sea ice loss within few years. Therefore, the wintertime SAT and zonal-mean zonal wind anomalies averaged during the last 150 years of model simulations are very similar to those averaged during the last 265 years (compare Supplementary Fig. 5 and Fig. 3). These results are consistent with previous studies (Olivié et al. 2012; Rugenstein et al. 2016) showing that the adjustment timescales of atmosphere and the ocean mixed layer to a radiative forcing given as a step function are only a few years. Therefore, discarding the first 50 years of model integration captures the equilibrium response of atmosphere whereas the deep ocean circulation takes several hundred years to reach an equilibrium.

We have now added the following text to the Methods section to clarify these points: "In our CESM1 simulations, the Arctic summer sea ice cover quickly responds to the mid-Holocene insolation forcing, within 10 years (Supplementary Fig. 5). The wintertime 200-hPa zonal-mean zonal winds averaged from 65°N to 80°N also responds to the Arctic sea ice loss within few years, although large internal variations of wintertime atmospheric circulations are overlaid the signal (Supplementary Fig. 5). Because of the relatively rapid adjustments of SIC and the associated atmospheric circulations, using the last 150 years of each simulation produces quantitatively similar results. For example, the wintertime SAT and zonal-mean zonal wind anomalies averaged during the last 150 years of model simulations are very similar to those averaged during the last 265 years (compare Supplementary Fig. 5 and Fig. 3). These results are generally consistent with previous studies showing that the adjustment timescales of atmosphere and the ocean mixed layer to a radiative forcing given as a step

function are only a few years^{39, 40}. Discarding the first 50 years of model integration therefore captures the equilibrium response of atmosphere, whereas the deep ocean circulation takes several hundred years to reach an equilibrium^{37, 39, 41} (Fig. 5a).”

Regarding the SSTs: we have an existing figure showing that the SST anomalies averaged over the last 150 years (Fig. 5g) are qualitatively similar to those averaged over the last 265 years (Fig. 5f).